# Phase Separation-Regulated Fungal Growth, Sexual Development, Adaptation and Synthetic Biology Applications

**DOI:** 10.3390/jof11090680

**Published:** 2025-09-17

**Authors:** Xinxin Tong, Daixi Zhang, Zhenhong Zhu

**Affiliations:** 1The Ministry of Education Key Laboratory of Standardization of Chinese Medicine, Key Laboratory of Systematic Research of Distinctive Chinese Medicine Resources in Southwest China, Resources Breeding Base of Co-Founded, School of Pharmacy, Chengdu University of Traditional Chinese Medicine, Chengdu 611137, China; zhangdaixi@cdutcm.edu.cn; 2School of Life Science, Zhejiang Chinese Medical University, Hangzhou 310053, China; zhenhongzhu@aliyun.com

**Keywords:** liquid–liquid phase separation, fungal biology, environmental cues, synthetic biology

## Abstract

Liquid–liquid phase separation (LLPS) is a fundamental biophysical process in which proteins and nucleic acids dynamically demix from the cellular milieu to form membraneless organelles (MLO) with liquid-like properties. Environmental cues, such as light, temperature fluctuations, and pathogen interactions, induce LLPS of fungal proteins with intrinsically disordered regions (IDRs) or multimerization domains, thereby regulating fungal hyphal growth, sexual reproduction, pathogenesis, and adaptation. Recently, LLPS has emerged as a powerful tool for biomolecular research, innovative biotechnological application, biosynthesis and metabolic engineering. This review focuses on the current advances in environmental cue-triggered fungal condensates assembled by LLPS, with a focus on their roles in regulating the fungal physical biology and cellular processes including transcription, RNA modification, translation, posttranslational modification process (PTM), transport, and stress response. It further discusses the strategies of engineering synthetic biomolecular condensates in microbial cell factories to enhance production and metabolic efficiency.

## 1. Introduction

Fungi can sense various environmental conditions for modulating physiological processes. Biomolecular condensates, also known as membraneless organelles (MLOs), can rapidly respond to sudden environmental changes, such as temperature, light, pH, starvation and energy depletion [1]. MOLs form through weak multivalent intermolecular interactions of protein [1]. MOLs were firstly observed over 100 years ago, are ubiquitous in eukaryotic cells, and play critical roles in diverse biological process. For example, in *Neurospora crassa*, live imaging of hyphae showed frequency clock protein (FRQ) foci characteristic of condensates near the nuclear periphery [2]. Further study showed that FRQ interacts with FRH (FRQ-interacting RNA helicase) and casein kinase I (CKI), forming a stable FRQ–FRH complex that represses its own expression by interacting with and promoting phosphorylation of its transcriptional activators white collar complex (WCC) [3]. In budding yeast, the essential translation initiation factor DEAD-box RNA helicase 1 protein (Ded1p) undergoes heat-induced gel-like condensates, revealing that Ded1p condensation selectively represses translation of housekeeping mRNAs to promote survival under severe heat stress [4]. In *Ashbya gossypii*, the RNA-binding protein Whi3 forms condensates throughout the cytoplasm around nuclei and at sites of polarized growth, facilitating hypha growth at certain temperatures [5]. In *Aspergillus oryzae*, poly(A)-binding protein (Pab1) accumulated as cytoplasmic foci mainly at the hyphal tip in response to stress [6]. Hence, response to diverse environmental stimuli, fungi form biomolecular condensates, facilitating stress response and regulating phycological process.

These biomolecular condensates exist in cytosolic compartments such as stress granules (SGs) and processing bodies (PBs) [7]. SGs composed of mRNAs and RNA binding proteins, and they assemble in response to stress-induced inactivation of translation initiation [7]. PBs are aggregates of translationally silenced messenger ribonucleoprotein particles (mRNPs) and mRNA degradation machinery components and share component with SGs, including sequester proteins involved in mRNA degradation [8]. Pab1in yeast, a defining marker of SGs, undergoes phase separation (PS) and forms hydrogels upon exposure to physiological stress such as a reduction in pH or heat shock in *S. cerevisiae* [9]. Prion-like low complex regions (LCRs) were first discovered in connection with prion-like proteins in budding yeast [10]. Under stress, the yeast prion protein Sup35 or its prion domain/prion domain and middle domain (Sup35N/NM) overproduce and form amyloid fibrils assemblies [11]. These biomolecular condensates triggered by various stresses participate in the basic cellular processes of fungi, including transcription, translation, posttranslational modification process (PTM), signaling, stress response, and active transport [1].

These biomolecular condensates exhibit liquid-like properties via liquid–liquid phase separation (LLPS), enabling their transition from a dispersed soluble state to a condensed phase distinct from their surrounding milieu [12] (Figure 1). Intracellular LLPS behaviors occur when biomacromolecules (e.g., proteins, nucleic acids) self-associate through multiple, weak interactions and reach a critical threshold concentration [12]. The multivalent interactions are enabled by repetitive amino acid motifs and structure flexibility within the intrinsically disordered regions (IDRs) and low-complexity regions (LCRs) [12]. To date, various computational methods have been developed to predict the LLPS propensity of proteins [13,14]. Given the significant biological roles of intracellular condensates, the artificial MOLs via LLPS was designed to produce proteins with novel functions, opening new avenues for synthetic biology applications [15].

LLPS serves as a fundamental mechanism that enables cells to integrate environmental into physiological process. However, reports on LLPS in fungi is relatively limited. In this review, we present an in-depth description of the molecular characteristics, biological functions, and mechanism of LLPS proteins in fungi, as well as the prevalent techniques for LLPS propensity prediction. Furthermore, we explore how the principle and strategies of biomolecular condensates are applied to synthetic biology.

## 2. LLPS in Fungal Photomorphogenesis, Hyphae Growth and Pathogenesis

### 2.1. Photomorphogenesis

Light controls important physiological and morphological responses in fungi. Photomorphogenesis is the process by which organisms adapt to light environment by adjusting their metabolism, growth, and development [16]. Photoreceptors sense light and regulate the expression and activity of core components of the molecular oscillator, thereby aligning internal rhythms with external light conditions [17]. The circadian clock is a conserved timekeeping mechanism that integrates biological process with environmental factors like light and temperature [17]. In *N. crassa*, the core clock protein FRQ undergoes LLPS to sequester FRH and CK1, thereby regulating CK1 activity and controlling WCC-mediated transcriptional suppression [3]. FRQ phosphorylation dynamically modulates its conformational flexibility and oligomeric state, with kinetics controlled by CK1 recruitment to IDRs and CK1 autoinhibition [18]. Furthermore, temperature-sensitive LLPS of FRQ creates a feedback loop that attenuates CK1 phosphorylation, revealing that PTM regulate FRQ’s physical properties via LLPS [2,18,19]. The clock also controls translation through rhythmic eIF2α phosphorylation, which sequesters specific mRNAs into circadian mRNPs during subjective daytime, effectively removing them from the translational pool [20,21] (Figure 2A). These findings highlight the pivotal role PS plays in regulating the photoperiod pathway, which integrates light signals with circadian clock-controlled processes.

White collar-1 (WC-1) functions as a central blue-light photoreceptor in fungi. It interacts with white collar-2 (WC-2) to form WCC, regulating the transcription of genes involved in early sexual development [22]. The regulatory activity is modulated by phytochromes (Phy-1 and Phy-2). Upon light activation, WCC initiates the expression of Vivid (VVD), which mediates photoadaptation by competitively forming WCC-VVD heterodimers, thereby effectively inactivating the photoresponse [23]. This negative feedback loop is further regulated by the transcriptional corepressor regulation of conidiation-1 (RCO-1)/its partner (RCM-1) complex, which accumulates in *Neurospora* nuclei and regulates *vvd* expression [24]. Notably, WC-1(Uniprot ID: Q01371) in *N. crassa* contains IDRs and LCRs, suggesting its propensity for undergoing PS, although experimental evidence remains elusive. Furthermore, in plants, the fungal Phy analogue PhyB forms subnuclear condensates via LLPS under red light, leading to its phosphorylation by photoregulatory protein kinases (PPKs) and subsequent degradation of phytochrome-interacting factors (PIFs) [25]. It was suggested that light may trigger LLPS in fungal photoreception systems, meriting further experimental investigation.

CRYs are blue-light photoreceptors structurally related to photolyases that play critical roles in plant responses to light [26]. Photoactivated CRYs and Phy rapidly translocate from the cytoplasm to the nucleus to form photobodies when exposed to light [27,28]. Beyond circadian regulation, blue light-induced the LLPS of CRY2/METTL16-type m^6^A writer FIONA1 (FIO1)/SUPPRESSOR of PHYTOCHROME A (SPA1) complex to form photobodies, dynamically regulating chlorophyll homeostasis and efficient photosynthesis in *Arabidopsis* [26,27]. These findings underscore that light-induced LLPS of photoreceptor/write complexes senses light signaling, integrating environmental cues to the developmental and metabolic processes in plant. So, RNA methylation-regulates PS, fine-tuning photomorphogenesis in plants. The orthologs of these photolyases, CRYs, and Phys have also been identified in fungi and have diverse functions, such as DNA repair, circadian clock resetting, and regulation of transcription [16]. Notably, these orthologs contain several LCRs and IDRs, i.e., Cry DASH (uniprot ID:Q4I1Q6, in *Gibberella zeae*, Q7SI68 in *N. crassa*), Phy attachment site domain-containing protein (uniprot ID: A0A1L9WHS0_ASPA1 in *Aspergillus aculeatus*), and Phy (uniprot ID: A0A2A9NZK7 in *Amanita thiersii*), suggesting their propensity for undergoing PS. However, light-induced LLPS of these photoreceptors has not been reported in fungi yet. Whether light may trigger LLPS in fungal photoreception systems and how it influences the fungal biology merit further experimental investigation.

### 2.2. Hyphal Growth and Pathogenesis

Fungal polarity is a sophisticated process that requires precise spatiotemporal organization and regulation of cellular machinery to enable polar growth and facilitate interspecies communication.

Whi3 functions as a key modulator of apical growth effectors, coordinating cell cycle events and morphogenesis. It is required for the apical localization of Cdc28-Cln1/2 complexes during bud morphogenesis, activating Cdc42 and its effectors at the growth bud tip [29]. In both *A. gossypii* and *S. cerevisiae*, Whi3 forms biomolecular condensates with the cell cycle regulator CLN3 mRNA [29,30]. These condensates localize near the nuclei [29,30]. This spatial organization further implies a potential role for Whi3 in stress-responsive RNA processing. In *Ashbya*, Whi3 constitutively assembles into functional condensates near nuclei and polarized growth sites [30]. Furthermore, Whi3 combined with SPA2 and Bni1 RNAs, and these condensates were required for promoting tip growth and new lateral branches [30] (Figure 2B). The biological functions of Whi3 condensates are determined by sequence variations within their glutamine-rich IDRs (QRR) [30]. A heptad repeat within the QRR was found to confer temperature-sensitive behavior [30]. It was demonstrated that variation in the size and composition of an IDR can drive cellular adaptation to specific temperature ranges. Aip5, a polarisome protein in budding yeast, synergizes with formin Bni1 to promote actin assembly [31]. Under stress, its N-terminal IDR mediates higher-order oligomerization and cytoplasmic condensate formation. These condensates are dynamically regulated by Spa2 via LLPS [32] (Figure 2C). So, Spa2-mediated biomolecular condensates contribute to actin polymerization and facilitate cellular adaptation to stress via LLPS [32]. Furthermore, in *A. oryzae*, AoPab1 undergoes dynamic resocialization and formed distinct cytoplasmic foci at hyphal tips under stress [6]. AoSO (SOFT protein), a homolog of *N. crassa* SO, is essential for hyphal fusion and colocalized with SGs in cells exposed to heat stress [6]. Collectively, PS integrates hyphal growth with transcriptional responses, finely tuning to cellular and environmental conditions.

Pathogens strategically utilize biomolecular condensates to alter membrane-bound organelles, escaping degradation and enhancing vesicle trafficking and reproduction in the host [33]. For example, *M. oryzae* undergoes a multistage morphological transformation during host plant infection [34]. The scaffolder protein MoSpa2 remodels actin cable networks in space and time by assembling the polarisome complex via the N-terminal IDRs, regulating hyphal growth and host infection [34]. Furthermore, condensates associated with pathogenesis dynamically assemble through PTMs (particularly phosphorylation and acetylation) in immune cell signaling, enabling quick and flexible responses during infection [35]. For example, the effector FolSvp2, secreted by the fungal pathogen *Fusarium oxysporum*, undergoes PS to form condensates in plant cells, hijacking a host protein (tomato iron-sulfur protein, SlISP) to suppress host reactive oxygen species (ROS) production and promote infection [36]. FolSvp2’s action needs K205 acetylation, which blocks ubiquitination-dependent degradation of this protein in both Fol and plant cells, suggesting that PTM regulates the assembly of these phase separated protein [36] (Figure 2D). Additionally, the apoplastic protein SlPR1 interacts with FolSvp2, inhibiting its entry into host cells and neutralizing its virulence effect [36]. These findings illustrate how pathogen effectors exploit PS to form functional condensates that manipulate host processes during infection.

## 3. LLPS in Cellular Processes in Response to Stresses

Biomolecular condensates are triggered by stresses such as heat stress, starvation, hypoxia, treatment with metabolic inhibitors, and other unfavorable conditions. LLPS has emerged as a crucial mechanism regulating the formation of biomolecular condensates, such as RNP or mRNP particles, SGs, and PBs, which is critical for modulating cellular processes.

### 3.1. RNA Processing

The assembly of large ribonucleoprotein SGs fall into a large class of protein- and RNA-rich cellular structures, including nucleoli, regulating translation. Several SGs-associated proteins were reported to be purified, such as FUS, Poly(A)-binding protein (Pab1), RNPA1/2, and Whi3, which form hydrogels in vitro upon exposure to physiological stress conditions [37]. SGs efficiently inhibit protein synthesis by sequestering mRNAs and translation factors.

Pab1 plays key roles in mRNA polyadenylation, stability, and translational control. Pab1 is consistently recruited to mRNA complex. Pab1 in yeast, a defining marker of SGs, phase separates and forms hydrogels in vitro upon exposure to physiological stress conditions, such as pH reduction or heat shock [38]. Pab1 consists of four RNA-recognition motifs (RRMs) that drive PS through electrostatic interactions, along with a proline-rich low-complexity region (LCR, P domain) enriched in non-aromatic hydrophobic residues that stabilizes the phase-separated state at elevated temperatures [38]. The heat-induced Pab1 condensates were dispersed by the Hsp40/Hsp70/Hsp104 disaggregation system, facilitating cellular recovery from heat shock [39]. The heat-induced PS of Pab1 exhibits exceptional temperature sensitivity, surpassing that of other known systems [38]. The Q10 of ~350 for the growth rate of Pab1 quinary assemblies at the onset of heat stress is remarkably higher than the Q10 of ~200 for the conductance change of heat-activated channel (*Anopheles gambiae* AgTRPA1) [38]. It was proposed that PS in environment-perceiving is highly efficient and widely utilized by cellular life.

Lsm7 is a component of the evolutionarily conserved Lsm1-7/Pat1 complex and actively contributes to SG assembly. It also co-localized with Pab1, promoting SG formation and stress adaptation in yeast [38]. Lsm7 undergoes LLPS through IDR and internal hydrophobic clusters [40]. Lsm7 PS condensates function as seeding scaffolds that promote Pab1 demixing and subsequent SG nucleation, likely mediated by RNA interactions [40]. This process presents a mechanism for SG formation under conditions of energy and nutrient limitation. Nevertheless, the specific components and signaling pathways regulating SG assembly under these or other cellular stresses remain to be fully understood. Furthermore, Lsm1 and Lsm4 of the Lsm1-7/Pat1 complex are localized to PBs in human cells [40]. In human cells, Lsm7 forms foci that contribute to SG assembly, providing valuable clues for understanding the mechanisms underlying SG formation and SG-associated human diseases (e.g., neurodegenerative diseases and cancer progression) [40].

Like Pab1, the poly(U)-binding protein (Pub1) is an essential component for SGs. Pub1 IDR undergoes condensates under starvation or heat stress, and this condensate formation is linked to cell-cycle arrest [39]. Purified Pub1 forms reversible, gel-like condensates under low pH conditions, while heat shock induces more solid-like structures, which mirrors in vivo observations and correlates with global translational inhibition and ribosome stalling [39]. Thus, different types of physiological stresses, as well as differences in stress duration and intensity, induce biomolecular condensates with distinct physical properties. These variations ultimately determine the distinct modes of stress adaptation and rates of cellular recovery.

Ded1p, an ATP-dependent DEAD-box RNA helicase, is one key component of yeast SGs. Ded1p undergoes PS in a manner strongly correlated with both the intensity and duration of a heat stress, rapidly forming condensates at temperatures exceeding 39 °C [39]. This mechanism selectively represses housekeeping mRNAs translation while promoting survival under conditions of severe heat stress. Under severe cellular stress, the liquidity of PBs is maintained by Hsp104, and loss of this activity results in PBs components entering SGs [41]. Orthologs of Ded1 across species, including mouse, *C. elegans* (LAF-1), and human (DDX3X), also contribute to the assembly of functional granules such as neuronal granules, SGs and PBs [42,43].

Collectively, SGs represent a conserved evolutionary strategy to cellular stress. However, SGs composition varies across organism reflect evolutionary divergence. In yeast, SG marker proteins such as Pub1, Pbp1, and Ded1p localize within SG. In human cells, SGs are primarily scaffolded by Ras GAP SH3 domain-binding protein (G3BP1/2), T-cell intra cellular antigen-1, and T-cell intracellular antigen-related protein, being associated with human pathophysiological processes [44,45]. The composition of SGs is dependent on the type of stress and the organism subjected to the stress. Research in yeast has provided insights into understanding SGs biology in and the mechanism in SG-associated diseases, including neurodegenerative disorder and cancer [46].

### 3.2. Translation

The translational repressor Scd6 and the decapping stimulator Edc3 function partially redundantly to promote PBs assembly in *S. cerevisiae* [47]. This is achieved by sequestering the Dcp1–Dcp2 decapping complex in the cytoplasm, thereby preventing its nuclear import mediated by the karyopherin β protein Kap95 [48]. mRNA decay factors (Dcp2, Pat1, and Edc3) cooperatively partition into PBs in *S. cerevisiae*, efficiently degrading cytoplasmic mRNA [47] (Figure 3A). Dcp2 undergoes multivalent interactions using short helical leucine-rich motifs (HLMs) in its disordered C-terminal domain [47]. The pathogenic yeast *C. albicans* dynamically regulates PBs formation in response to external signals, facilitating rapid adaptation [49]. The key PBs components, Dcp2, Dhh1, Edc3, and Kem1/Xrn1, exhibit stress-dependent sequestration into PBs under various growth conditions [50]. Notably, Edc3 are crucial for the assembly and/or maintenance of PBs and in the hyphal morphogenesis [50]. Localization studies demonstrated that the core components of PBs included the decapping machinery (Dcp2 and Dhh1), 5′–3′ exoribonuclease (Kem1/Xrn1), and Edc3 [50]. The disordered segments in Dcp2 and FET proteins tend to adopt compact conformations, which is necessary for PS [51,52]. The evolutionary conservation may help proteins preserve the capability to undergo PS. PBs are uniquely enriched with factors related to mRNA degradation and decay.

The essential translation factor Sup35 forms protective gels via pH-induced LLPS as a response to stress [53]. The PS of Sup35 occurs independently of the N-proximal prion-like domain (PrLDs) that contain LCR and a pH sensor domain [54] (Figure 3B). Sup35 IDR mediates the formation of reversible, functional gels during stress, enabling rapid translation recovery post-stress [53] (Figure 3B). Under pH stress, Sup45 undergoes PS, forming dense condensates along with Sup45p and Pab1p [54]. Notably, most prion-like domains (PLDs, *S. cerevisiae* contains >200 proteins with such domains) likely function to modulate PS or protein solubility rather than to forms prions [54]. Thus, prion-like domains represent conserved environment stress sensors that facilitate rapid adaptation in unstable environments by modifying protein phase behavior. PrLDs display intrinsic disorder and exhibit LLPS behavior, driving diverse biological functions [55]. These domains possess distinctive sequence grammars that determine their PS behaviors across eukaryotes [55]. For example, early flowering 3 (ELF3) acts as a thermosensor through PrLD-mediated LLPS in the thermal induction of flowering in *Arabidopsis* [56]. PrLDs within IDR of ARID1A drive its LLPS, promoting oncogenic progression in Ewing’s sarcoma [57]. In budding yeast, Ded1p undergoes heat-induced PS into gel-like condensates [4] (Figure 3C). This adaptive response represses housekeeping mRNA translation and promotes stress mRNA translation [4].

In response to stress, biomolecular condensation through PS modulates gene expression and translation, thereby enabling precise spatiotemporal regulation of adaptive functions [7,54]. This mechanism appears to be evolutionarily conserved across eukaryotes, highlighting its fundamental role in cellular stress adaptation and survival [38,54]. The expansion of IDRs in higher eukaryotes likely facilitated the emergence of more sophisticated condensation mediated regulatory networks, whose dysregulation is implicated in human diseases [7].

### 3.3. Chromatin Organization

LLPS also plays a crucial role in chromatin organization, as exemplified by heterochromatin foci, where the heterochromatin protein 1a (HP1α) undergoes PS to maintain transcriptional repression [58]. Chromatin tethering to the nuclear periphery is important for resisting nuclear deformations in response to microtubule polymerization-based pushing forces in fission yeast [59]. HP1α forms phase-separated heterochromatin domains that maintain nuclear stiffness and resist mechanical deformation [58]. This process is regulated by non-coding RNA (ncRNA)-dependent Cryptic loci regulator 4(Clr4) self-association, which promotes H3K9me2 deposition, while ubiquitin-conjugating enzyme4—Cryptic Loci Regulator complex (Ubc4-CLRC) and ubiquitin-conjugating enzyme3 (Ubp3) dynamically modulate centromeric transcription via ubiquitination [60]. HP1 is an evolutionarily conserved chromatin protein found from fission yeast to mammals [58]. While fission yeast and Drosophila HP1 undergo LLPS and promote heterochromatin clustering, mammalian HP1 has lost this ability, reflecting an evolutionary attenuation of LLPS capacity [61]. This finding suggests that PS can be finetuned across evolution. Furthermore, Clr4 engages with nucleosomes through its chromodomain and disordered regions to promote de novo methylation and di- but not tri-methylation [62]. PS also governs epigenetic regulation of chromatin. For example, the polycomb repressive complex 1 (PRC1) bind H3K27me3-marked heterochromatin via Chromobox 2 (CBX2), which undergoes LLPS to mediate chromatin compaction [63]. PS acts as a conserved regulatory principle governing epigenetic regulation of chromatin in fungi.

Switch/sucrose nonfermentable chromatin-remodeling complex (SWI/SNF) remodelers alter histone-DNA interactions and are important for regulating chromatin architecture and gene expression [22]. The SNF5 subunit of the yeast SWI/SNF complex is glutamine- and asparagine-rich LCR that undergoes pH-responsive PS [64]. This behavior suggests its LCRs function as a direct pH sensor, potentially enabling rapid modulation of transcriptional activity in response to acid stress [64]. This observation supports an emerging paradigm that LCR-mediated PS equips key regulatory proteins, particularly those involved in the processes of transcription and translation, to rapidly respond to environmental stress. Furthermore, some subunits of human canonical BRG1/BRM-associated factor (cBAF) complexes contain increased intrinsic disorder relative to those of yeast SWI/SNF complexes, controlling condensate formation and heterotypic interactions [65]. The evolutionary expansion of IDRs in higher eukaryotes likely enabled more sophisticated transcriptional regulation through molecular condensation, a mechanism whose dysregulation contributes to diseases such as cancer and neurodevelopmental disorders.

### 3.4. Other Cellular Processes

In the budding yeast *S. cerevisiae*, the minimum transport machinery consists of the membrane proteins peroxisomal 13 (Pex13) and Pex14 and the cargo-protein-binding transport receptor [66]. In peroxisomal biogenesis, Pex5 and Pex13 undergo PS to facilitate cargo import [66]. IDRs in Pex13 and Pex5 resemble those found in nuclear pore complex proteins [66].

Fluorescence imaging shows that cargo import is linked to temporary clustering of GFP-Pex13 and GFP-Pex14 on the peroxisome membrane [66]. Pex13 and Pex14 form foci at different saturating content, suggesting they may create channels by LLPS of Pex5–cargo with Pex13 and Pex14 [66]. Thus, LLPS enables the formation of transient protein transport channels [67]. Additionally, Pex5, Pex13, and Pex14 exhibit functional and mechanistic conservation across eukaryotes [66]. Defects in this system in humans lead to peroxisome biogenesis disorders [67]. Notably, although the IDRs within these proteins display considerable sequence variation, none of these variants are associated with disease [68]. This may be explained by the robustness of PS to sequence variations within IDRs [68].

In *F. graminearum*, the epigenetic regulator the bromo-adjacent homology (BAH)-plant homeodomain (PHD) domain containing protein BAH–PHD protein 1 (BP1), containing two IDRs, forms nuclear condensates through its IDR2 domain through LLPS [69]. This process facilitates the recruitment of the polycomb repressive complex 2 (PRC2) and promotes trimethylation of histone H3 at lysine 27 (H3K27me3) [69]. So, PS play a role in transcriptional silencing of genes involved in secondary metabolism [69]. Furthermore, PRPP amidotransferase (PPAT) reversibly localizes to cytoplasmic condensates upon the depletion of extracellular purine bases in budding yeast [70]. Pyruvate kinase Cdc19 reversibly aggregates into protective foci upon glucose starvation and heat shock in budding yeast [71].

A summary of the biomolecular condensates in fungi discussed above, along with their respective biochemical mechanisms and physiological outcomes, is shown in Table 1.

## 4. Identification of Intrinsically Disordered Proteins/Regions

Multivalent proteins often undergo PS through interaction between LCDs, IDRs, prion-like domains (PrLDs), RNA-binding domains (RBDs), which are typically enriched in charged, polar, and aromatic residues [55]. IDRs, characterized by their structural plasticity, drive PS via van der Waals forces, electrostatic attraction, π–π stacking, and cation–π interactions, which promote the nucleation and growth of biomolecular condensates [72], while their PS propensity is further modulated by amino acid composition (e.g., Arg/Gly-rich RGG/RG motifs or Gln/Asn-enriched PrLDs), charge distribution, hydrophobicity, and PTM [1]. RBDs exhibit modular architectures in which IDRs are coupled with IDRs (e.g., RGG/RG repeats), enabling multivalent RNA binding to amplify PS [73]. PrLDs, characterized by LCRs with Gln/Asn enrichment, rely on Tyr/Arg-mediated interaction via aromatic–aromatic stacking and aromatic–arginine interaction in RNA binding domains to display protein multivalency [74]. Interactions between these residues drive the PS of FET family proteins, including FUS, EWS, and TAF15, playing crucial roles in mRNA processing, transcriptional regulation, and DNA repair [51]. The FUS LCR with low-complexity amyloid-like reversible kinked segments (LARKS) enhances significant condensate densification [51].

To date, various tools for predicting PS proteins have been developed and several databases have been released. Early predictors generally based on specific protein features, such as PLAAC, PAPA, and PrionScan, can be used to identify prion-like proteins by comparing the amino acid similarity of protein candidates to that of *bona fide* yeast prion [75,76]. Later, more powerful methods for PSP prediction have been developed, such as ParSe, Droppler, LLPhyScore, LLPSDB, PSPredictor, PSPer, PhaSePred, MambaPhase, MolPhase, and PhaseHub [77,78,79,80,81,82]. These methods adopt machine learning techniques and use different sequence features (e.g., IDR, pi interaction, PLD, LCR, NCPR, FCR, hydrophobicity, and Shannon Entropy) and training datasets [83]. The sequence features associated with LLPS are shown in Figure 4. A list of common prediction tools of PS propensity is shown in Appendix A. For instance, WC-1 forms a complex with WC-2, known as WCC, which functions as a central regulator of circadian rhythms and photoresponses across diverse fungal species [3]. Previous studies have demonstrated that WCC binds to two light-responsive elements (LREs) in *frq* promoter to mediate both circadian and light-dependent gene expression [3,84,85]. Here, the online LLPS prediction tools (ParSe, PSPredictor, PSP hunter and MolPhase) were employed to analyze the LLPS propensity of WC-1 (UniProt ID:001371). As a result, all algorithms consistently predicted a high LLPS propensity of WC-1 (score approaching 1.0), with values exceeding the 0.90 threshold considered indicative of extremely high likelihood to undergo PS (Appendix A). The MolPhase prediction analysis is shown in Figure 5. These computational results suggest that LLPS could serve as a novel and crucial regulatory mechanism for WC-1-mediated photo-sense and -response in fungi, which remains to be clarified and is being validated in our study.

## 5. LLPS in Synthetic Biology

De novo designed proteins undergo LLPS to form biomolecular condensates and artificial organelles. These structures provide precise spatial and temporal control over cellular processes. LLPS has been designed to construct artificial organelles that compartmentalize metabolic pathways. This spatial organization enhances enzymatic reaction kinetics and enables the creation of responsive systems within living cells. This approach is revolutionizing metabolic engineering and expanding the capabilities of synthetic biology.

### 5.1. Engineering Strategies of Protein Condensates

Approximately 30% of the eukaryotic proteome are IDPs [86], exhibiting unique structural plasticity that facilitates dynamic macromolecular interactions and self-assembly [87]. Yifan Dai et al. designed synthetic IDPs using octapeptide repeats (G-R-G-D-S-P-Y-S)_n_ (n = 20–80), where amino acid substitutions enabled precise control condensate saturation (nM-mM) and PS behavior [88]. Artificial IDPs with varied molecular weights and aromatic contents yield modulable PS in vitro and in vivo [89]. These condensates can sequester enzymes and adjust catalytic efficiency based on IDP size, providing a versatile LLPS platform for spatial control of cellular functions [89].

IDPPs exhibit remarkable environmental response and have four types of distinct LLPS in response to temperature changes: lower critical solution temperature (LCST), upper critical solution temperature (UCST), and combined “hourglass” or “closed loop” models [90]. ELRs typically exhibit UCST behavior [90]. ELRs can be precisely designed with transition temperatures below physiological level, enabling their use in injectable drug delivery that form gradual-release depots through temperature-induced LLPS [90]. Fatty acid-modified ELR was produced through one-pot recombinant expression and post-translational lipidation in *E. coli*, efficiently forming lipid–polymer hybrids that combine the thermoresponsive behavior of ELRs with the self-assembly properties of fatty acids. Using pH-responsive ELRs, sequential compartmentalization can be achieved within protocell systems through controlled H^+^ influx. This process drives coacervation by modulating the protonation states of IDPPs and altering the local micropolarity inside condensates, enabling dynamic spatial organization in synthetic biology and biomaterial applications [91]. Resilin-like polymers (RLRs) typically exhibit UCST behavior in response to a decrease in temperature. RLRs can concentrate ions and form interfacial electrical double layers that exhibit redox-active microenvironments [92]. Expression of these polypeptides in *E. coli* modulates ion distribution and membrane potential, improving uptake of charged molecules [92].

FUS LC domains exhibit intrinsic self-association properties, forming dynamic biomolecular condensates. This is employed to induce PS by fusion to target proteins. PTMs regulate the domain’s N-terminal (NTC) and C-terminal (CTC) cross-β core, enabling selective expansion of NTC- or CTC-mediated polymerization [93]. N-terminal acetylation (Nt-Ac) of FUS LC enhances its PS propensity while reduces aggregation, revealing a critical role of Nt-Ac in regulating physiological LLPS and preventing pathological aggregation [94]. FUS N fusions with the Cry2 enhances light-induced PS in the optoDroplet/optoCluster systems [93]. These light-responsive systems enable spatial and temporal control of PS, increasing product yields via compartmentalization of intermediate metabolites [93].

Coiled-coil (CC) motifs (30–40 aa) are ubiquitous in fibrous proteins and transcription factors and undergo tunable post-translational self-assembly into higher-order structures [95]. Their interaction strength and specificity can be engineered via sequence and length variation, making them ideal modular tags for protein assembly [96]. For instance, a 34-residue CC peptide assembles into pH-dependent structures: fibrils at acidic pH and spherical oligomers under neutral conditions [95].

The adaptor protein Nck drives PS via its multivalent interactions. Its three SH3 domains bind proline-rich motifs in N-Wiskott–Aldrich syndrome protein (N-WASP), while its SH2 domain engages phosphorylated tyrosine residues on nephrin, forming dynamic condensates that coordinate actin remodeling [97,98]. Min Liu et al. identified a phosphotyrosine (pY)-containing peptide derived from nephrin/Tir similar sequence, promoted Nck/N-WASP LLPS, functionally mimicking the role of phosphorylated nephrin [97], while peptide p1, a covalent blocker targeting the Nck-SH2 domain at Lys^331^, was designed to effectively disrupts condensates formation and prevents bacterial infection in intestinal epithelial cells [97].

Small ubiquitin-related modifier (SUMO) belongs to the ubiquitin-related protein family [99]. Recent works have engineered biomolecular condensates using SUMO-interacting motif (SIM)/SUMO interactions system, where small ubiquitin-like modifier proteins and their recognition motifs enable programmable assembly of functional condensates [99]. The IDRs proteins/regions with engineering potential are listed in Table 2, highlighting the manipulation of PS in synthetic biology.

### 5.2. Applications of LLPS in Synthetic Biology

Using IDR tags or interaction peptide tags, diverse biomolecules of interest can be incorporated into phase-separated proteins, enabling the creation of functional condensates with novel function. In *Bacillus subtilis*, a “stacking blocks” strategy was developed for de novo design of an 8-mer FW1 (RGYGSPDG) and a series of synthetic IDPs [100]. This approach enables the assembly of multifunctional, micron-scale condensates through targeted molecular localization, resulting in 2.23-fold increase in 2′-fucosyllactose (2′-FL), a fourfold increase in the translation specificity of engineered enzymes, and an increase of 75.0% in *N*-acetylmannosamine [100]. This system provides a foundation for engineering condensates with tailored multifunctional function. To address the leakage of toxic or volatile intermediates that impairs the synthesis efficiency in microbial cell, an artificial MLO platform was constructed by using engineered IDPs in *S. cerevisiae*, featuring a tunable regulation platform for spatiotemporal control over MLO size and rigidity [15]. As a result, methanol and malate assimilation increased by 162% and 61% through size modulation, respectively, while acetyl-CoA synthesis switched from oxidative to non-oxidative glycolysis via rigidity modulation, reducing CO emissions by 35% and increasing n-butanol yield by 20% [15]. This artificial MLO platform provides a versatile and efficient strategy for enzyme co-localization, enabling channeling of C1 substrates into valuable biochemicals [15] (Figure 6). The short peptides RIAD and RIDD are derived from AMP-dependent protein kinase (PKA) and the A kinase-anchoring proteins (AKAPs), respectively [101]. Due to their small size, strong binding affinity, and 2:1 binding stoichiometry resulting in branched architectures, they were used create scaffold-free enzyme assemblies [101]. In vitro, enzymes in the menaquinone biosynthetic pathway were assembled through the short peptide tags, and nanoparticles were obtained with various stoichiometries, sizes, geometries, and catalytic efficiency [101]. In *E. coli*, assembling the last enzyme of the upstream mevalonate pathway with the first enzyme of the downstream carotenoid pathway a synthetic metabolic node led to the formation of a pathway node, increasing carotenoid production by 5.7-fold [101]. The same strategy enhanced lycopene yield by 58% in *S. cerevisiae* [101]. The work presents a simple strategy to impose metabolic control in biosynthetic microbe factories.

To further expand the toolbox, Kejin et al. presented an optogenetically controlled PS system that precisely regulates biomolecular condensate dynamics through blue light-controlled tobacco etch virus (TEV) protease and subsequent cleavage of IDPs in *S. cerevisiae* [102]. This light-responsive system significantly increased squalene and UA yields by 32.4% and 46.4%, respectively, demonstrating a robust optogenetic strategy for controlling biomolecular condensate dynamics to optimize biosynthetic efficiency [102].

Spider silk proteins have been engineered to form protein condensates that spatially organize enzymes to recruit a variety of enzymes to improve the efficiency of enzyme catalysis by using the interaction of short peptides or direct fusion based on the silk protein [103]. This approach was applied to the de novo synthesis of 2′-FL [103]. The pathway was modularly optimized and co-localized within the synthetic compartments to form multienzyme aggregates, significantly increasing 2′-FL titers compared to WT and free enzyme systems [103]. These spider silk-driven condensates improve metabolic flux and reduce intermediate diffusion, offering a programmable and scalable platform for constructing synthetic metabolic modules and optimizing bioproduction in microbial cells.

Collectively, LLPS has emerged as a powerful tool in synthetic biology. In eukaryotes, membrane-bound organelles create isolated microenvironments that can be functionally reprogrammed via targeted recruitment of biomolecules. In contrast, prokaryotes lack such endogenous organelle systems, limiting the implementation of spatial organization strategies. Encapsulating enzymatic reactions within native organelles enhances reaction kinetics for natural product synthesis. However, de novo design and functionalization of genetically encoded proteins that form cellular condensates for engineering cellular metabolism or protein translation remains an ongoing challenge.

## 6. Conclusions and Future Perspectives

LLPS has emerged as a fundamental cellular mechanism that modulates diverse biological processes in fungi, including hyphal growth, sexual development, pathogenesis and stress adaption, and essential cellular processes, such as gene transcription, DNA repair, cell homeostasis maintenance, signal transduction, and chromatin remodeling. LLPS serves as a universal adaptative strategy for fungi to adapt to environmental such as temperature, pH, and nutrient starvation. The evolutionarily conserved process is primarily driven by multivalent interactions via IDRs, which facilitate the dynamic formation of biomolecular condensates with physicochemical activities. The variation in IDRs across eukaryotes indicates that they regulate sophisticate cellular activities via LLPS. This is particularly prominent in higher eukaryotes, and the dysregulation of LLPS is implicated in numerous human diseases. Although the PS mechanism of these condensates has been investigated and revealed, the experimental conditions often fail to accurately mimic those inside living cells in vitro. The physiological relevance of these findings remains unclear. A major unanswered challenge is whether and how PS of these RNA-binding proteins directly contributes to an organism’s cope with stress.

In recent years, significant advances have been made in the de novo design of scaffold proteins capable of forming synthetic MLOs both in vitro and in vivo, either constitutively or in response to stimuli. Future advances in synthetic condensates will enable precisely programmable control of cellular functions and signaling pathways through quantitative analysis, standardized modular design, and precise spatiotemporal regulation of synthetic condensates.

## Figures and Tables

**Figure 1 jof-11-00680-f001:**
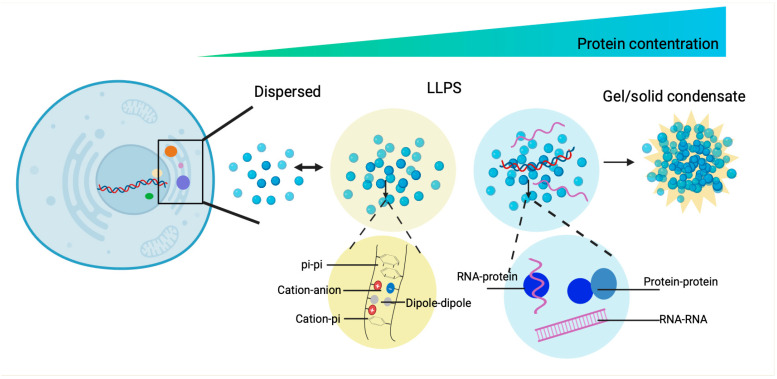
The dynamic spectrum of material states of phase separated condensates and the driving forces of phase separation. The biomolecular condensates formed through LLPS are highly dynamic and exchange components with their surroundings. With the increase in protein concentration, the condensates of liquid-like condensates turn irreversibly to gels or amyloid fibrils. These condensates are stabilized by diverse multivalent interactions, including π-π stacking of aromatic residues, cation-π and electrostatic interactions between charged side chains, dipole–dipole alignment, and conventional binding motifs between proteins and proteins and proteins and nucleic acids.

**Figure 2 jof-11-00680-f002:**
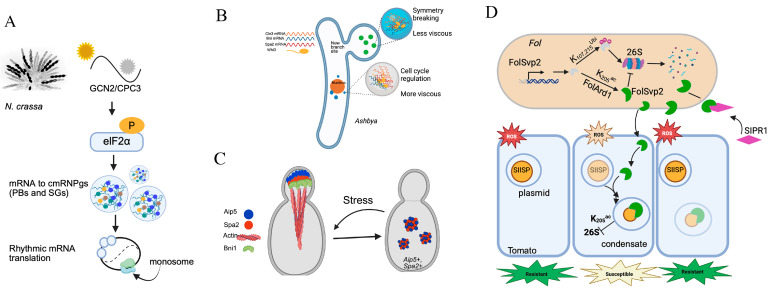
LLPS regulates the fungal photomorphogenesis, polarized growth, and pathogenesis. (**A**). The circadian clock controls the kinase GCN2/CPC-3, which rhythmically phosphorylates eIF2α. In fungi, this periodically suppresses translation of certain mRNAs and promotes their sequestration into P-bodies, resulting in rhythmic translation. (**B**). PS in polarized growth, cell asymmetry, and nuclear divisions in *A. gossypii*. Whi3/Bni1/Spa2 mRNA droplets locally nucleate actin to initiate symmetry breaking at branching sites, while Whi3/Cln3 mRNA droplets form adjacent to the nucleus where they regulate asynchronous nuclear division. (**C**). Spa2 localizes Aip5 to the bud tip, where it nucleates actin filaments via recruiting Bni1. Under stresses (e.g., low pH or energy depletion), Aip5 and Spa5 form reversible cytoplasmic condensates. (**D**). The fungal effector FolSvp2, secreted by *F. oxysporum*, undergoes PS and relocates the tomato protein SlISP from plastids into condensates within plant cells. This process suppresses ROS production and facilitates fungal invasion. FolSvp2 function depends on acetylation at K205, which prevents its ubiquitination and degradation. In response, tomato produces the apoplastic protein SlPR1, which binds FolSvp2, blocks its entry into host cells, and neutralizes its virulence.

**Figure 3 jof-11-00680-f003:**
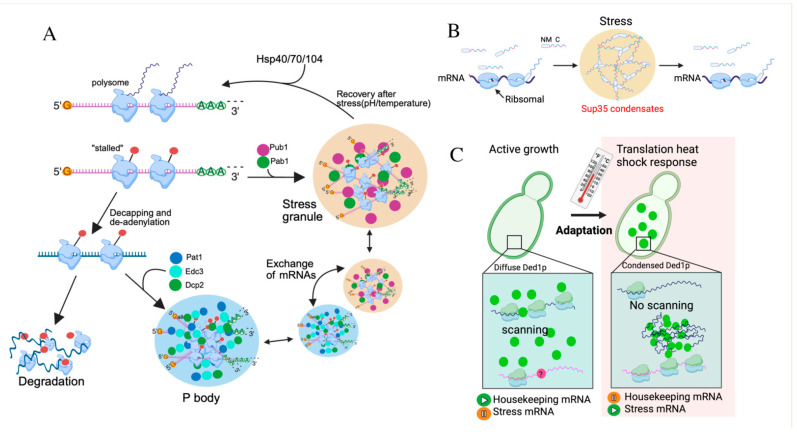
LLPS dynamically regulates the fundamental cellular processes in fungi. (**A**). Environmental stressors like heat shock or pH shifts trigger rapid translation inhibition and ribosome stalling, diverting mRNAs to PBs or SGs. PH stress induced SCs to dissolve spontaneously, whereas heat-induced SGs require chaperones (Hsp40/Hsp70/Hsp104). Subsets of mRNAs that are concentrated in PBs can either be degraded or exchanged with SGs. (**B**). Sup35 exhibits PS regulated by its N-terminal prion domain during cellular stress. Sup35 consists of a disordered prion domain (green), a stress-sensing module (red), and a folded C-terminal catalytic domain (blue). Under cellular stress, the prion domain and the sensor domain interact to promote the assembly of protective and reversible biomolecular condensates. (**C**). The model of Ded1p PS promotes a translational switch from housekeeping to stress protein production. Under normal conditions, the RNA helicase Ded1p promotes translation of housekeeping mRNAs by unwinding their 5′ UTR structures. During heat stress, Ded1p undergoes PS and incorporates into SGs. consequently, Ded1p-dependent housekeeping transcripts are silenced, whereas stress mRNAs are translated. Silencing mRNAs require Ded1p together with other unidentified cofactors. ‘?’: unidentified cofactors.

**Figure 4 jof-11-00680-f004:**
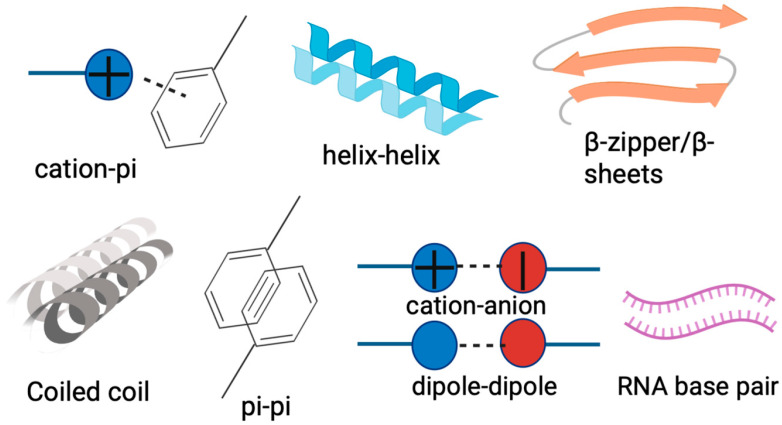
The sequence features aiding in predicting LLPS propensity proteins.

**Figure 5 jof-11-00680-f005:**
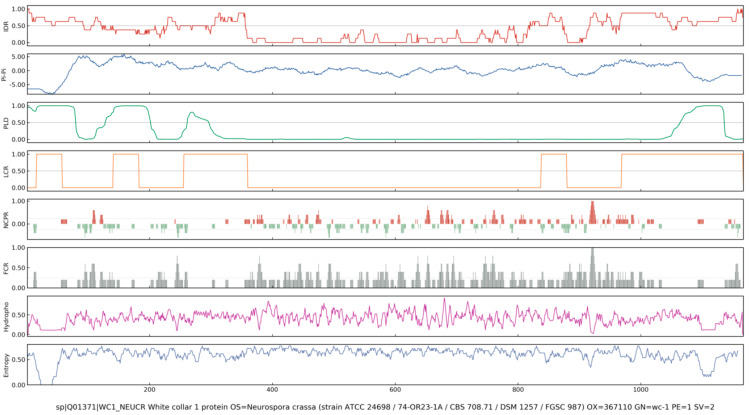
The MolPhase prediction analysis of WC-1 (UniProt ID:001371). Hierarchy of predictive features aiding in predicting potential PS proteins. Features, from top to bottom, are IDR, pi interaction, PLD, LCR, NCPR, FCR, hydrophobicity, and Shannon Entropy. IDR: Intrinsically disordered regions, pi interaction: π-π interaction potential, PLD: Prion-like domains, LCR: Low-complexity regions, NCPR: Net charge per residue, FCR: Fraction of charged residues.

**Figure 6 jof-11-00680-f006:**
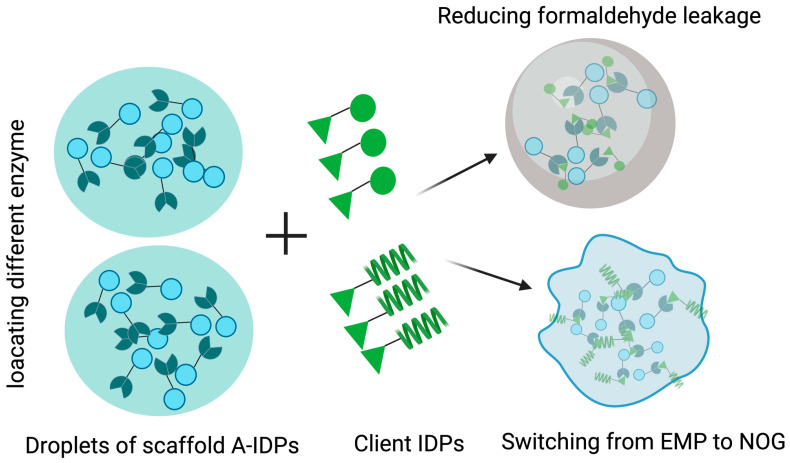
An artificial MLO was engineered to enhance the biosynthesis in *S. cerevisiae*. Through precisely adjusting the size and rigidity of these MLOs, toxic formaldehyde accumulation was reduced, and metabolic flux was redirected from oxidative to non-oxidative glycolysis. Consequently, both the methanol assimilation efficiency and n-butanol production significantly increased. EMP: Embden–Meyerhoff–Paranas, NOG: Non-oxidative Glycolysis, A-IDP: Artificial Intrinsically Disordered Protein.

**Table 1 jof-11-00680-t001:** Summary of biomolecular condensates in fungi discussed in this review, along with their respective biochemical mechanisms and physiological outcomes. WCC: White Collar Complex, FRQ: clock gene frequency protein, FRH: FRQ-interacting RNA helicase, CK1: Casein Kinase 1, QRR: glutamine-rich region, eIF2α: eukaryotic translation initiation factor 2α, PTM: post-translational modification, Fol: *Fusarium oxysporum* f. sp. lycopersici, ROS: reactive oxygen species, SlISP: iron-sulfur protein, RBPs: RNA binding proteins, H3K27me3: trimethylation of lysine 27 in histone H3, Cdc19: yeast PK, PPAT: PRPP amidotransferase, Pex: Peroxisomal proteins, BP1: Bromo-adjacent homology-plant homeodomain domain containing protein 1, IDR2: intrinsically disordered region 2 of BP1, ‘-’: undefined.

Assembly Proteins	Species	LLPS Region	Biological Mechanism	Stress	Refs.
FRQ/FRH/CK1	*N. crassa*	IDR	modulates CK1 activity, coordinates repression of WCC, FRQ phosphorylation enhances conformational flexibility and alters oligomeric state	light	[2,3]
eIF2α	*N. crassa*	-	PTM-mediated circadian rhythms and stress tolerance	light	[20,21]
MoSpa2	*M. oryzae*	N-terminal IDRs	polarized actin cable assembly	pathogenesis	[34]
FolSvp2	*F. oxysporum*	-	translocates SlISP from plastids into effector condensates in planta, attenuates host ROS production to facilitates the fungal invasion	pathogenesis	[36]
Whi3/*SPA2/BNI1 RNAs*	*S. cerevisiae*	QRR	maintains tip growth and initiates lateral branching	heat stress	[29,30]
Whi3/*CLN3*	*S. cerevisiae* *Ashbya*	QRR	temperature adaptation	cold stress	[29,30]
Aip5/Bni134/Spa2	*S. cerevisiae*, and filamentous fungi	N-terminal domain	protects actin assembly	pH or energy depletion response	[31]
Scd6/Dcp1/2/Pat1/Edc3	*S. pombe*	HLMs	mRNA storage and decay	-	[47,48]
Lsm7 foci	*S. cerevisiae*	IDR	mRNA storage and decay	conditions of energy and nutrient limitation	[40]
Sup35	*S. cerevisiae*	PrLDs	rescuing essential Sup35 translation factor from stress-induced damage	pH stress response	[11,53]
Snf5p	*S. cerevisiae*	-	transcription and chromatin remodeling	pH stress	[64]
Pub1	*S. cerevisiae*	RRMs drive self-assembly while IDRs modify condensate properties	helps cells recover from heat shock	heat shock/pH stress	[39]
Pab1	*S. cerevisiae*	LCR	-	heat stress	[38]
Ded1p	*S. pombe*	IDR	enhances survival	heat stress	[4]
BP1	*N. crassa* *F. graminearum*.	DR2	regulates BP1–PRC2 interaction and H3K27me3 recognition to repress secondary metabolism-related genes expression, particularly those involved in deoxynivalenol mycotoxin biosynthesis	-	[69]
Pex5/Pex13/Pex14	*S. cerevisiae*	IDR	forms minimal transport machinery on nuclear pore and peroxisome membrane	-	[66,67]
Ubc4-CLRC	*S. cerevisiae*	IDR	regulates centromeric transcription; chromodomain function via ubiquitination	-	[60]
HP1α/Swi6	*S. pombe*	multivalent interactions of the N-terminus and hinge region	promotes H3K9me2 deposition, modulates nuclear stiffness	-	[58]
Cdc19	*S. cerevisiae*	IDR	protects Cdc19 from stress-induced degradation and inactivates enzymes	glucose starvation and heat shock stress response	[71]
PPAT	*S. cerevisiae*	-	purine synthesis	purine-depleted environment	[70]

**Table 2 jof-11-00680-t002:** The list of intrinsically disordered proteins/regions. IDPs: intrinsically disordered proteins, LARKS: Low-complexity aromatic-rich kinked segments, CC: coiled coil, FUS: fused in sarcoma, LC: low complexity, RLR: Resilin-like polypeptides UCST: SH3: SRC-homology 3 domains, SUMO: Small Ubiquitin-like Modifier, ELP: elastin-like polypeptides, SIM: SUMO-interacting motif, ‘-’: undefined.

Motif.	Sequence	Nanostructure/Assembly	Refs.
IDPs	an octapeptide repeats of (G-R-G-D-S-P-Y-S)	macroscopic PS	[86,89]
ELR, LCST like IDPPs	(VPGXG)n, (IPGXG)n, (VPAXG)n, (VPAPVG)n X can be any amino acid except L-proline (P).	coacervate/spherical micelles/micellar aggregates	[90,91]
RLR, UCST like IDPPs	(GRGDSPYS)20 (RDGSPSS-GRGDYPYS)_10_ (GGRPSDSXGAPGGGN)_n_ X must be an aromatic amino acid such as tryptophan (W) or phenylalanine (F), in the case of tyrosine (Y)	nanofibrillar assemblies	[92]
FUS LC domain	(G/S-Y-G/S)_27_	a gel-like state, β amyloid-like polymers, fibrils	[94]
LARKS	_58_ NFGAFS_63_	amyloid-like protofilaments, hydrogels	[51]
CC	heptad repeat, usually containing hydrophobic amino acids at the first and fourth position of the repeat.	fiber-forming	[95,96]
Nck	SH3 domains	-	[97,98]
SUMO-SIM	an extended β-strand-like conformation	-	[99]

## Data Availability

The original contributions presented in this study are included in the article/Appendix A. Further inquiries can be directed to the corresponding author.

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
