# Peer review of "Phase Separation-Regulated Fungal Growth, Sexual Development, Adaptation and Synthetic Biology Applications"

_jof, 2025, doi:10.3390/jof11090680_

Round 1
Reviewer 1 Report
Liquid-liquid phase separation (LLPS) is a fundamental cellular mechanism enabling the formation of biomolecular condensates, which organize cellular components without membrane boundaries. The article offers a comprehensive overview of the role of LLPS in fungal biology and its applications in synthetic biology.
The article is interesting and informative, but it contains several inaccuracies that require clarification.
See attached file

Reviewer 2 Report
I have reviewed the manuscript entitled “Phase Separation-Regulated Fungal Growth, Sexual Development, Adaptation and Synthetic Biology Applications.” The topic is timely and relevant, as liquid–liquid phase separation (LLPS) has become an emerging area of study in fungal biology and synthetic biology. While the manuscript provides a comprehensive collection of recent studies on fungal LLPS, the current version reads more like an encyclopedic compilation rather than a critical synthesis. Many sections summarize findings at length without clearly evaluating their broader significance, limitations, or unresolved questions. For example, plant photobody studies are discussed in detail, but their relevance to fungi is not critically assessed, and the distinction between well-established fungal LLPS mechanisms versus speculative parallels is often blurred. The authors should refine the narrative to highlight key conceptual advances in fungal LLPS, explicitly contrast fungal systems with other eukaryotes, and identify areas where mechanistic evidence remains incomplete. A stronger emphasis on synthesis—rather than extensive cataloging—would substantially improve the clarity, focus, and impact of the review.
1. Define abbreviations at first use consistently (e.g., IDR, LLPS, PTM)
2. Correct typos, e.g., “morhphogenesis” → “morphogenesis”
3. The manuscript requires substantial language polishing. There are many grammatical errors, awkward phrasing, and inconsistencies in tense and style (e.g., “fungi forms biomolecular condensates” should be “fungi form”; “LLPS severs” should be “LLPS serves”). A thorough English revision is strongly recommended.
4. The synthetic biology section is promising but disproportionately long relative to the fungal biology sections. Since this maunscript is intended for Journal of Fungi, more emphasis should remain on fungal LLPS mechanisms, with the synthetic biology discussion condensed and focused on fungal applications.
5. Figures (e.g., Fig. 2, Fig. 3) are dense and text-heavy. They should be simplified to convey key concepts more visually. Table 1 is comprehensive but could be reformatted for readability (e.g., clearer column structure, consistent abbreviations)
Round 2
Reviewer 2 Report
the revised mauscript looks good in shape and all my previous comments have been resolved.
the revised mauscript looks good in shape and all my previous comments have been resolved.